# Souvenirs Development Related to Cultural Heritage: A Thematic Review

**Qiuxia Zhu *** , **Rizal Rahman, Hassan Alli and Raja Ahmad Azmeer Raja Ahmad Effendi**

Faculty of Design and Architecture, University Putra Malaysia, Serdang 43400, Selangor, Malaysia
* Correspondence: qxzhu826413@gmail.com

**Abstract:** Previous souvenir research has generally supported the use of heritage arts and crafts as a design inspiration, which both arouses visitors' interest in the destination and enhances the local cultural identity. The increasing concern about souvenirs and cultural heritage has led to a proliferation of studies on this topic. However, few review articles discuss their inevitable connection and the process of transforming cultural heritage resources into souvenirs. Therefore, this review aims to analyze the literature on souvenirs related to cultural heritage from 2018 to July 2022. A criterion for inclusion and exclusion is presented to search and screen articles from three leading databases (i.e., WOS, Mendeley, and Scopus). Ultimately, 27 articles meeting the eligibility criteria were selected for synthesis analysis. ATLAS.ti 9, as the analysis software, yielded quantitative results showing trends in research on cultural heritage-related souvenirs. At the same time, from the qualitative analysis emerged four main themes about the topic: (1) significance of souvenirs, (2) customer's purchase intention, (3) transformation, and (4) sustainability. Finally, the findings are expected to be beneficial in addressing the relationship between souvenirs and cultural heritage and provide insights for future research and practice of cultural product sustainability.

**Keywords:** souvenir; cultural heritage; transformation; purchase intention; sustainability; ATLAS.ti

## 1. Introduction

The importance of cultural tourism is reflected in the economic flows and other potential areas of destination cities, whose image, prestige, and even the new cultural and social activities generated by these new flows of tourism are influenced. This flow has now expanded from large monumental cities in the conventional sense to several cities with World Heritage Sites [1]. Culture consists of people's attitudes, beliefs, ideas, values, and what they do regarding their conventional patterns of conduct or style of life, which may be seen in their artifacts, artworks, and cultural creations [2]. Consequently, Carson et al. [3] argue that developing heritage items for travel consumption is a strategy to increase tourist interest to encourage travel to less popular destinations.

Tangible products sold in the context of tourism are used as examples of indigenous cultural heritage that is intangible and represents native religion or spirituality. They are often seen as problematic cases of commercialization [4]. Heritage values are the result of the interaction between cultural and environmental systems. However, Olalere [5] considers that the changes in lifestyles brought about by the expansion of globalization have led to a decline in traditional cultural aspects, resulting in many cultural objects being seen as primitive and outdated. Hence, the in-depth examination of souvenirs supports the use of regional arts and crafts to reflect the cultural heritage and teach young people about their cultural heritage, enabling them to define local in their own ways [5,6].

## 2. Literature Review and Research Aim

### 2.1. The Role of Cultural Heritage

In inheritance and identity, the term cultural heritage can be given multiple implications. Cultural resources can be considered priceless presents passed down from one

generation to the next and survive as enduring reminders of a transforming national identity [7]. The view of Ashworth [8] on the relationship between tourism, cultural heritage, and the economy shows that cultural heritage as an industry is a modern practice that can be controlled and organized to make marketable goods. At the same time, the consensus in Europe is to integrate a site's cultural heritage with its monuments and museums, which are seen as the undisputed keepers [6].

The problem is how the significance of ancient relics and objects is distributed in modern society in the context of tourism. According to Evans-Pritchard [9], historical sites can be economically directly "recycled" while being utilized socially to reinforce national ideology. They are kept as pieces of an irretrievable past. As he puts it, nostalgia is a "strong marketing weapon for selling history" and an emotional experience linked to one's sense of the past. The cultural heritage industry's image of presenting its culture and history through national monuments is an essential factor that distinguishes it from other countries [6]. For instance, Mohale et al. [10] concluded that even though not enough focus has been attached to cultural and heritage tourism in some parts of South Africa, the evidence suggests that the sector is creating both local economic growth and an entrepreneurial spirit. In a similar case in northern Japan, the indigenous Ainu people are adept at decorating their clothing, houses, utensils, ornaments, and spiritual objects with several distinctive motifs. Today, visitors from Japan and abroad appreciated souvenirs and artifacts with these motifs, making them valuable from both cultural and economic points of view [11]. Overall, these cases support the view that by integrating traditional art into the design of contemporary cultural and creative products, folk art can be preserved and help people comprehend folk culture's meaning and aesthetic value [12]. Therefore, it should be cultivated by the authorities to enhance the niche potential of the destination tourism economy.

### 2.2. Commercialization of Cultural Heritage

Because of the non-renewable value of cultural heritage, governments and the arts community are treating ancient objects as commodities to promote destination development potential. Certainly, commercialization does not always change the locality's unique identity but is one of the main tools to highlight it [6]. Husa [13] considers the commercialization and commodification of the souvenir producers' own material culture, also referred to as "self-commodification," primarily determined by the sense of identity of particular ethnic groups, which in turn leads to their different responses to the commercialization process. Since culture is a critical factor in designing acceptable products in contemporary society, cultural value is the core of products. Consequently, design can be a motivating force for cultural development [5,14]. Typically, souvenir suppliers consider handcrafted items, patterns and presentations, and skilled craftsmanship necessary to produce authentic souvenirs at their destinations. They recognize the importance of creating souvenirs to reflect the cultural and historical features and their origins. The human element of souvenir creation emphasizes the handcrafted value. Therefore, local artisans and craftspeople are crucial to creating ethnic and traditional items representing their unique heritage, culture, and lifestyle [15].

### 2.3. Preservation and Conservation

Ahmad [16] found that some of the fascinating natural and cultural items mentioned in the study need to be preserved and conserved to make them more viable and appealing. Scholars have proposed a new strategy for cultural heritage development, namely "cultural heritage rejuvenation," to solve the dilemma of cultural heritage preservation and development [17]. In the same vein, Bullen and Love [18] in their article also note that a preferable strategy for achieving conservation is heritage revitalization. This view is supported by Wu et al. [19], who writes that combining tourism-related activities with craft culture could preserve craft culture. The museum beside the Grand Canal in Hangzhou, China, in the article, is exemplified and explains creatively preserving craft traditions by encouraging contact between visitors and artisans.

*2.4. Relationship between the Souvenirs and Cultural Heritage*

A fresh appreciation of the value of traditional culture has been sparked by the shifting functional characteristics of cultural resources [20]. Manola and Balermpas [6] believe that souvenirs have an essential role in tourism and cultural heritage, and cultural heritage, specifically, can be served to tourists in the form of events and experience-based performances and tangible souvenirs. Thus, an authentic souvenir is a complex experience that combines the memory of travel with the artistic and cultural heritage of the destination. Meanwhile, products that seek to have a unique and attractive presentation require the specific skills of local artisans or the involvement of identifiable craftspeople [15].

Aboriginal people in some countries have begun to market tourism services related to regional crafts, music, and celebrations to express their ideas and spirits [21]. Ahmad [16] believes that the resource potential of some cultural heritage sites has not been exploited, and instead enhancing, protecting, and preserving this enriched heritage and transforming it into tourism products can go a long way in improving the quality of life of native communities. All of the responses made by the participants from a study in response to a survey on the value of local handicrafts as heritage assets for the sustainability of cultural tourism indicate that various tourist products in varied ways of crafts to present culture [22]. Therefore, designers need to consider cultural factors in the development of new products because it is through exploring and understanding these cultural factors and translating them into appropriate product design features that traditional products have the potential to gain new value [23].

Souvenirs are charming little pieces that encompass and illuminate the global–local interface of tourism. We must be aware of their complexity as they involve individuals, places, and implications [24]. Souvenirs are typically items visitors buy while on a trip in popular culture. They are an essential component of the travel experience and can also be as trip markers or mementos to preserve memories and serve as tangible reminders of their travels. Souvenirs are, therefore, more than just a remembrance of the past; they also serve as a lens to observe the culture of a place [6,13]. As a result, suppliers seek out, collect and analyze the unique local character of a destination and then pack them into a commodity [15].

There is a general perception that tourism, as one of the most critical sectors of the global economic field, combines a range of business practices to develop a well-rounded "product" that is traded in domestic and international environments. Manola and Balermpas [6] studied the interest of tourists in buying souvenirs and the significance of cultural heritage, revealing a dynamic relationship between tourism products and national identity icons. Souvenirs are part of a series of decisions taken by national authorities to emphasize specific elements of their cultural heritage.

The purpose of the souvenir is related to the extensive narrative [4]. Those souvenirs with indigenous spiritual meaning are a reminder of the traveler's visit. In a case study of Mambang pottery in Malaysia, Olalere [5] suggests that commercializing indigenous crafts can help improve visitor experiences. This strategy is beneficial to preserve heritage values while promoting cultural value to visitors. Furthermore, the production and sale of souvenirs can advantage local communities economically.

To sum up, undoubtedly, selling souvenirs brings economic benefits and creates more space for intangible cultural heritage to survive. While it benefits local communities economically, it can also lead to the degradation of natural landscapes and the commodification of native culture [25]. Furthermore, there is a gap between highlighting specific features of a destination's identity and creating stereotypes [6]. Liu et al. [26] propose that since intangible cultural heritage is detached from the daily life of present-day individuals, its original functions are no longer needed by the user. Thus, in souvenir business promotion, it makes sense to try to capitalize on tourists' interest in national tourism and encourage them to bring home traditional culture; it is also worth noting that original traditions cannot be confused with cheap copycats, and they need to be better-preserved [27].

In brief, with the growth of cultural tourism, the link between souvenirs and cultural heritage is getting increasingly relevant. It would appear that industry practice in this area is outpacing theoretical research. The growing number of publications focus on addressing souvenirs and cultural heritage from cultural studies and tourism management. However, there has been little discussion about the review analysis of this topic, so much so that it is impossible to understand its current trends and future research directions.

Therefore, there is a need to employ systematic reviews of the literature discussing souvenirs related to cultural heritage from the year 2018 to 2022 through the following research question: what are the current trends on souvenirs related to cultural heritage discussed in the literature from 2018 to 2022?

*2.5. Research Aim*

This paper aims to identify trends in souvenirs linked to cultural heritage and their interrelationships by reviewing the literature on souvenirs from 2018 to July 2022, focusing on the cultural heritage characteristics of souvenirs. The study results are expected to provide insights into the sustainability of souvenirs. In addition, this article will lay the groundwork for future research on the relationship between souvenirs and cultural heritage.

To achieve these objectives, this paper is divided into four parts. The first part deals with the current research on souvenirs linked to cultural heritage and then presents the research questions. The second part concerns the methodology, data collection, and analysis procedures used for this study. In turn, part three analyses the results by employing the quantitative and qualitative methods of twenty-seven selected articles, focusing on the four key themes that are (1) significance of souvenirs, (2) customer purchase intention, (3) transformation, and (4) sustainability. The following part four derives a framework for the role of cultural heritage resources in souvenir implementation. The final section presents the conclusions of this research and recommends further investigation.

**3. Materials and Methods**

Clarke and Braun [28] proposed the definition of thematic analysis as a process of identifying patterns and constructing themes through an in-depth reading of the subject matter, which contributes to understanding research trends in souvenirs related to cultural heritage. The use of thematic analysis in this study is a suitable approach to identify the new theory and concept for a less-established field. The benefit of thematic analysis is that it is an accessible, flexible, and valuable research tool, which has the potential to contribute a detailed and informative, yet elaborate, description of the data and to offer terms and "recipes" for thematic analysis in a theoretically and methodologically rational way [28,29].

The goal of this review is to analyze and explain the current literature overview of souvenirs related to cultural heritage; however, the topic of souvenirs and cultural heritage has only gained attention in recent years, and there are limited studies related to the transformation of cultural heritage resources into souvenirs and sustainability. Therefore, the thematic review is prepared according to the procedure introduced by Zairul [30] to capture those crucial data related to the research question through themes, which stand for some degree of patterned response or meaning in the data sets [29,30] This study is dedicated to analyzing and interpreting the findings and suggesting recommendations for future grounded theory on souvenir-related cultural heritage. The literature was selected based on the following criteria: (1) published in 2018–2022, (2) having at least 'souvenir' or 'cultural product' as a keyword, and (3) associating souvenirs with cultural heritage. A systematic methodological framework is adopted for this study from formulating the research question, selecting the data source, retrieval, and pre-processing; then extracting, analyzing, and synthesizing the themes; and, finally, visualizing and presenting, interpreting, and discussing the results.

The literature search was conducted in three databases of Web of Science, Mendeley and SCOPUS. According to the censored exclusion and inclusion criteria set, the initial search resulted in 160 articles from Web of Science, 49 results from Mendeley, and 42 ar-

ticles from SCOPUS. However, there are 36 papers to be down due to previous results or inconsistency with the topic, and some of them are incomplete or inaccessible with a fragmented link. In addition, there are several duplications in metadata (*n* = 4). Five results are moved since the literature is limited to English. The papers to be reviewed were dropped to 27 results and uploaded as primary files into ATLAS. Ti 9. Each article was then classified into author, volume and issue number, periodical, publisher, and publication year for further analysis (Table 1 and Figure 1).

**Table 1.** Search strings.

| Database | Search Strings | Results |
|---|---|---|
| Web of Science | TITLE: (souvenir) OR ("cultural product") and "cultural heritage" and English (Language) and Articles OR Editorial Materials OR Review Articles (Document Types) and Hospitality Leisure Sport Tourism or Humanities Multidisciplinary or Art or Cultural Studies (Web of Science Category) Timespan: 2018–2022 | 160 results |
| Mendeley | "Cultural heritage" and "souvenir" AND DOCUMENT TYPE: Article YEAR: [2018 TO 2022] | 49 results |
| SCOPUS | TITLE-ABS-KEY (souvenir) OR TITLE-ABS-KEY ("cultural product") AND TITLE–ABS–KEY (cultural AND heritage) AND LANGUAGE (English)) AND PUBYEAR > 2017 AND LIMIT-TO (DOCTYPE, "ar") AND LIMIT-TO (SUBJAREA, "SOCI") OR LIMIT-TO (SUBJAREA, "ARTS") | 42 results |

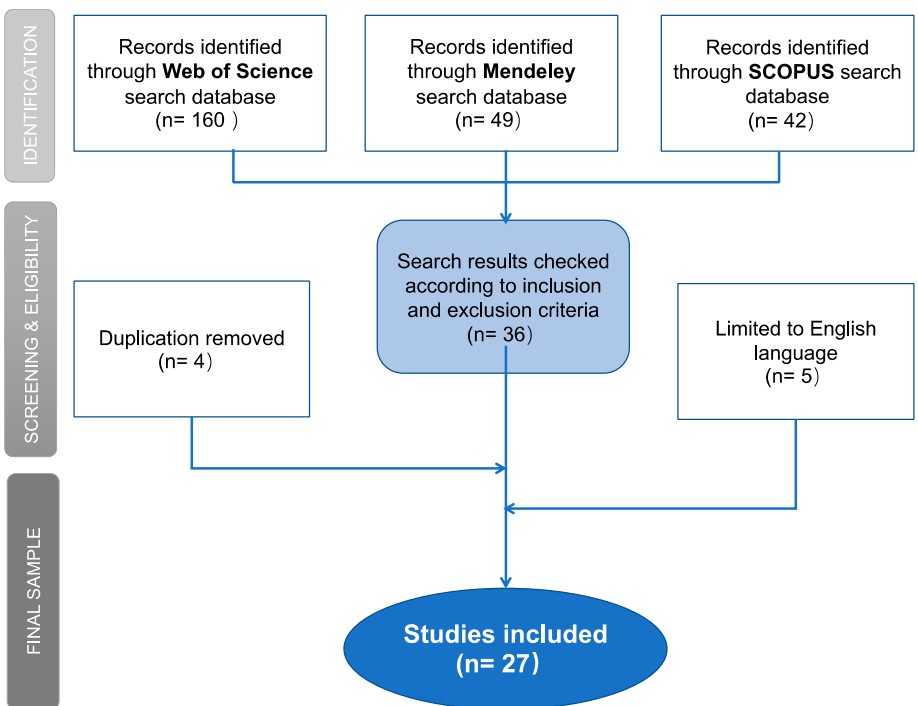

**Figure 1.** Inclusion and exclusion processes in the thematic review.

The articles were afterward assessed using both quantitative and qualitative analysis methods. The quantitative section reported the findings from a mathematical of view to derive the respective data. At the same time, the qualitative area extracted codes from the selected papers, inducting themes and developing a conceptual framework.

## 4. Results and Discussion

The significant findings of the thematic review are presented in this part. Quantitative and qualitative analyses were used to assess the selected 27 articles for answering the research question.

### 4.1. Quantitative Findings

Research trends in souvenir-related cultural heritage can be partially reflected by analyzing word frequency, the year of publication, the research location, the source of publication, and the theme. First, the quantitative section generated the following word cloud based on the analysis of the source documents (Figure 2). As shown in Figure 2, the most famous words that appeared in the cloud were "Tourism"," Cultural", "culture", "Heritage", "Value", and "Tourists", indicating their high word frequency in the article. As previously mentioned, this article focuses on souvenirs related to cultural heritage. The word cloud shows the main terms in this topic, with the word "Tourism" being mentioned 1656 times, followed by "Cultural" and "Culture" respectively 1440 and 540 times, while "Heritage", "Value", and "Tourists" were referred to 1062, 733, and 711 times.

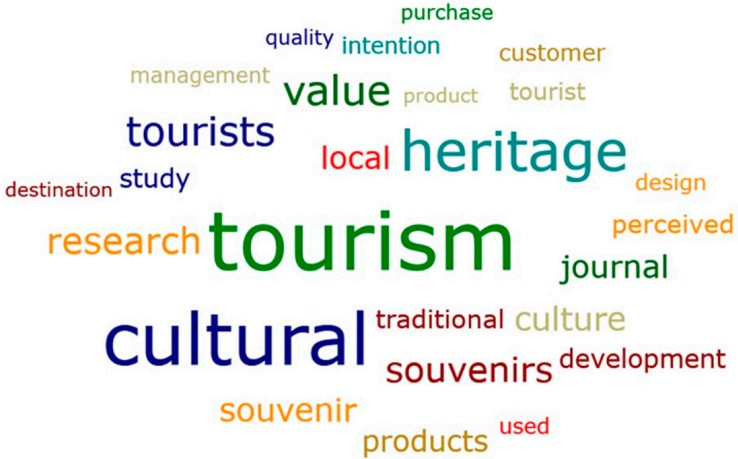

**Figure 2.** Word cloud generated from 27 articles.

Figure 3 illustrates the number of publications related to this topic, with three articles in 2018, four in 2019, nine in 2020, seven in 2021, and four in 2022. The overall trend is on the upward side about this topic, but there is a slight drop after peaking in 2020. The tendency may be due to the shocks of the COVID-19 pandemic on tourism, as it might have led to a change in research focus.

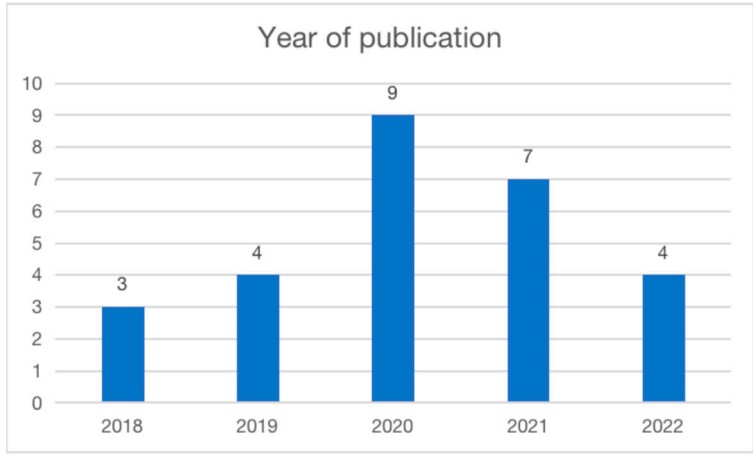

**Figure 3.** Year of publication.

The results obtained from the preliminary analysis of the geographical dispersion are set out in Figure 4. Regarding the number of articles, the topic of souvenirs and cultural heritage is more prevalent in Asia, especially in China. For instance, Sun et al. [31] examine the process of transforming Chinese "ritual cultural features" into "modern product forms". Liu [32] explores the relationship between product involvement, perceived value dimensions, and purchase intention of intangible cultural heritage (ICH) souvenirs. Scholars have also investigated the production and transmission of culture in museums from the perspective of social practices [19]. In addition, two articles were contributed from Turkey, Norway, South Africa, Malaysia, and Japan, respectively. These studies present an overview of the research, with Özgit et al. [22] from Turkey exploring residents' perspectives on the sustainability of cultural resources. Tashi and Ullah [11], on the other hand, specify how the symmetrical patterns of the Ainu heritage in Japan were used as inspiration for the virtual and physical prototyping. Similar explorations have come from academics in other regions, such as Malaysia, South Africa, and Norway. Patterns regarding regional distribution suggest that research is focused on countries with rich cultural heritage resources advocating a function for souvenirs in cultural heritage tourism and providing opportunities for communities.

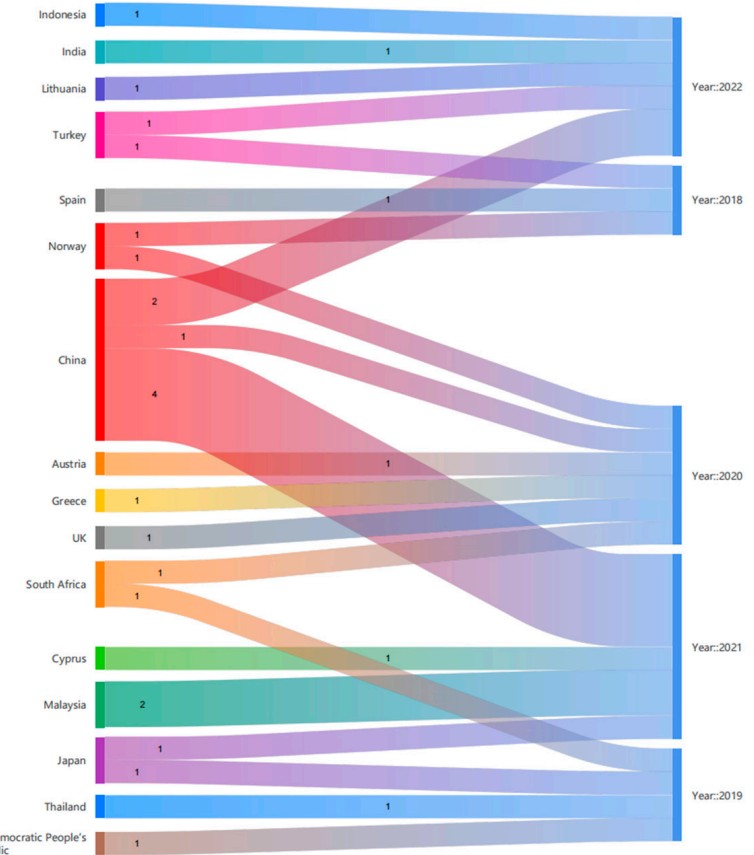

**Figure 4.** Country of studies and years of publication with the number of articles.

Table 2 below presents the selected subject trends and patterns for the publications. There were initially twelve coded attributes, but after renaming and merging, the coding results were reduced to four themes: significance of souvenirs, customer's purchase intention, transformation, and sustainability, which will be analyzed in detail later in the qualitative section. Many of the studies in the selected articles focus on the process of transforming cultural heritage into souvenirs. Consumer purchasing behavior is also a vital concern of the research, and the themes of meaning and sustainability of souvenirs have increased in the last two years (Table 3).

**Table 2.** The theme according to year.

|  | 2018 | 2019 | 2020 | 2021 | 2022 | Totals |
|---|---|---|---|---|---|---|
| Transformation | 2 | 2 | 3 | 6 | 3 | 16 |
| Customer's purchase intention | 1 | 1 | 2 | 1 | 2 | 7 |
| Significance of souvenirs | 0 | 1 | 2 | 1 | 1 | 5 |
| Sustainability | 0 | 0 | 0 | 2 | 2 | 4 |
| Totals | 3 | 4 | 7 | 10 | 8 | 32 |

**Table 3.** Documents to a theme table.

| | Significance of Souvenirs | Customer's Purchase Intention | Transformation | Sustainability |
|---|---|---|---|---|
| Ahmad (2021) [16] | ✓ | | ✓ | |
| Manola and Balermpas (2020) [6] | ✓ | | | |
| Berjozkina and Karami (2021) [33] | | | ✓ | ✓ |
| Husa (2020) [13] | ✓ | | ✓ | |
| Jerlei (2022) [34] | ✓ | | ✓ | |
| Köse and Akbulut (2018) [35] | | | ✓ | |
| Liang and Qi (2021) [12] | | | ✓ | |
| Liu (2021) [32] | | ✓ | | |
| Liu et al., (2019) [26] | | ✓ | | |
| Liu et al., (2020) [36] | | ✓ | | |
| Lopez-Guzman et al., (2018) [1] | | ✓ | | |
| Ma et al., (2021) [20] | | | | ✓ |
| Mathisen (2020) [4] | | | ✓ | |
| Mayuzumi (2022) [27] | | ✓ | | ✓ |
| Mohale et al., (2020) [10] | | | ✓ | |
| Olalere (2019) [5] | | | ✓ | |
| Özgit et al., (2022) [22] | | | | ✓ |
| Rahman et al., (2021) [37] | | | ✓ | |
| Sawagvudcharee et al., (2020) [38] | | ✓ | | |
| Schilar and Keskitalo (2018) [24] | | | ✓ | |
| Sharma (2022) [7] | | | ✓ | |
| Soukhathammavong and Park (2019) [15] | ✓ | | | |
| Sun et al., (2022) [31] | | | ✓ | |
| Tashi and Ullah (2019) [11] | | | ✓ | |
| Wu et al., (2021) [19] | | | ✓ | |
| Yotsumoto and Vafadari (2021) [25] | | | ✓ | |
| Zhou and Pu (2022) [17] | | ✓ | | |

The analysis of published sources shows that journals in the tourism management and culture categories are favored by the choice of souvenir and cultural heritage researchers. As shown in Table 4, *Religions*, *African Journal of Hospitality, Tourism, and Leisure*, and *Journal of Heritage Tourism* are the top three popular options for souvenir researchers. As mentioned earlier, if only "souvenirs" or "cultural products" were used as keywords for this search, the number of articles found turned out to be in the thousands. However, the findings displayed a noticeable decrease after adding cultural heritage to the search string. They were more centralized, proving that the topic is still fresh and could be explored more in the future.

**Table 4.** Articles reviewed based on journal.

|  | 2018 | 2019 | 2020 | 2021 | 2022 |
|---|---|---|---|---|---|
| *African Journal of Hospitality, Tourism and Leisure* |  | 1 | 1 |  |  |
| *American Journal of Industrial and Business Management* |  |  |  | 1 |  |
| *Annals of Tourism Research* |  |  | 1 |  |  |
| *Archives of Business Research* |  |  | 1 |  |  |
| *Asia Pacific Journal of Tourism Research* |  |  |  |  | 1 |
| *Asia-Pacific Journal of Regional Science* |  |  |  | 1 |  |
| *International Journal of Design in Society* | 1 |  |  |  |  |
| *International Journal of Early Childhood Special Education* |  |  |  | 1 |  |
| *International Journal of Economics, Business and Management Research* |  |  | 1 |  |  |
| *Journal of Cultural Heritage Management and Sustainable Development* | 1 |  |  |  |  |
| *Journal of Heritage Tourism* |  |  | 2 |  |  |
| *Journal of Material Culture* |  |  |  | 1 |  |
| *Journal of Service Science and Management* |  |  | 1 |  |  |
| *Journal of Sustainable Tourism* |  |  | 1 |  |  |
| *Journal of Tourism and Cultural Change* |  |  |  |  | 1 |
| *Modern Economy* |  | 1 |  |  |  |
| *Religions* |  |  |  | 1 | 2 |
| *SMART MOVES JOURNAL IJELLH* |  |  |  |  | 1 |
| *Symmetry* |  | 1 |  |  |  |
| *Tourism Culture and Communication* | 1 |  |  |  |  |
| *Tourism Management* |  | 1 |  |  |  |
| *Webology* |  |  |  | 1 |  |
| *WORLDWIDE HOSPITALITY AND TOURISM THEMES* |  |  |  | 1 |  |

Overall, this section provides an understanding of research trends in cultural heritage-related souvenirs through quantitative results that reflect, to some extent, the possibilities for product development in the literature. Although these studies have addressed numerous aspects, there is a difference in thinking between commercial strategies and the preservation of traditions [27], and few studies explicitly state the relationship between souvenirs and cultural heritage. Although local craft products are part of the cultural heritage, policymakers and planners do not pay sufficient attention to them [22]. As in most tourism sectors, the cultural heritage tourism sector also faces intense competition, making it particularly urgent to study the impact of the quality of cultural heritage revival experiences on revisit intentions in this context [17].

*4.2. Qualitative Results*

This section is a qualitative analysis that explains the themes derived from answering the research questions after reviewing relevant articles. The theme and direction of the relationship between cultural heritage and souvenirs were first coded. Afterward, the coding was synthesized and inducted to identify theories and concepts extensively considered and studied by the researcher. Four main themes were eventually identified: (1) significance of souvenirs, (2) customer's purchase intention, (3) transformation, and (4) sustainability. These themes do not exist independently but may overlap between articles, and thus some articles may employ several themes simultaneously. The following section will discuss each theme in depth, citing results outside the articles reviewed as required to answer the research questions and develop a conceptual framework for transforming cultural heritage resources into souvenirs (Figure 5).

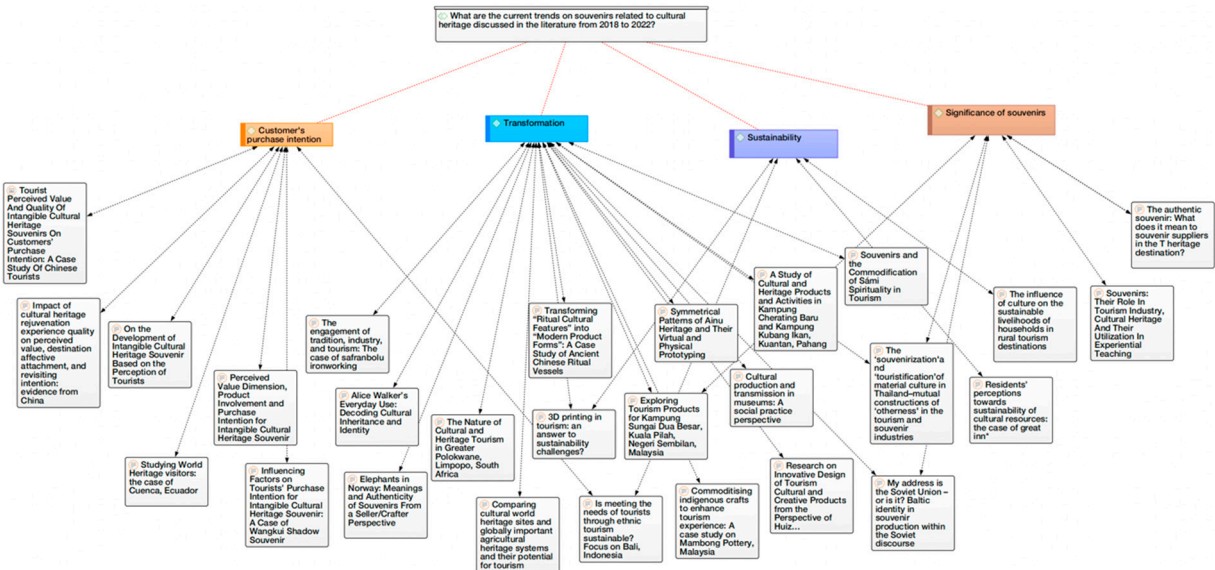

**Figure 5.** Overall network.

RQ: What are the current trends on souvenirs related to cultural heritage discussed in the literature from 2018 to 2022?

### 4.2.1. Theme 1: Significance of Souvenirs

Souvenirs combine the memory of travel with the artistic and cultural heritage of the destination and comprise a complex travel experience. To be precise, souvenirs not only stand for the identity and image of a particular culture but are also signs of history, cultural heritage, and geography [6]. For example, research from Jerlei [34] supports the use of the old Baltic towns as a repetitive motif on souvenirs whose European identity can be confirmed to differentiate the Baltic countries from the rest of the Soviet Union Nevertheless, from the perspective of some souvenir suppliers, souvenirs authentically reflect the local World Heritage Site. They can also extend the meaning and significance of a national cultural and historical asset to a broader region. The consensus among suppliers on the origin and locality of souvenirs focuses on the solid pursuit of objective authenticity. However, the challenge is the increasing demand for local production of souvenirs and the fierce competition between local markets and international traders [15]. Therefore, there is a need to release the potential of souvenirs to create space to assess and transform them as a medium that allows them to observe and understand the past and the present while shaping the future [6] (Figure 6).

### 4.2.2. Theme 2: Customer's Purchase Intention

Tourism with deep or superficial cultural motivations has proliferated in recent decades and has become a tool for regional development in the socio-economic sphere [1]. Husa [13], considers the need to examine how consumers and producers construct and produce national images through souvenirs. Based on gaze theory [39] the authors investigate the inter-construction of the "other" in the Thai tourism and souvenir industries, i.e., the process of "souvenirization" and "tourismization" of physical culture. The study demonstrates the impact of tourists on the "typical" image of handicraft production and souvenir business in Thailand since the 1960s, describing how visitors and host communities can mutually construct and reproduce their respective cultural 'others.' Liu [32] finds that the perceived value factor had a significant positive effect on the purchase intention of ICH souvenirs, while product involvement moderated the impact between the perceived value factor and purchase intention in another empirical study. The study also revealed that the souvenir production sector should pay regard to tourists' perceived value when designing

and marketing ICH souvenirs, as cultural heritage attitudes have a significant positive impact on perceived value and customers' purchase intention to engage in souvenirs.

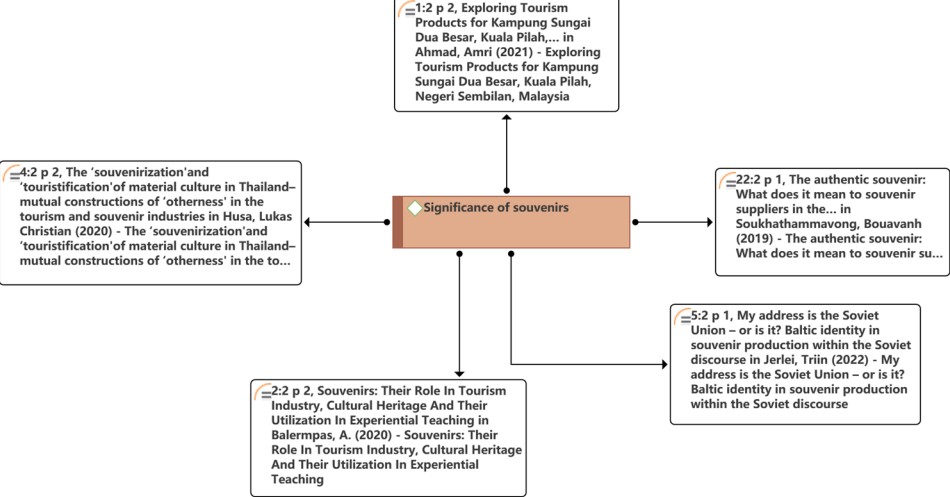

**Figure 6.** Network on the significance of souvenirs [6,13,15,16,34].

One of the essential elements in the development of World Heritage Sites is studying the link between motivation and satisfaction. The study results of Lopez-Guzman et al. [1] indicate the presence of three dimensions of visitor motivation: circumstantial, cultural, and convenience. Likewise, according to the motivation context, tourists can be classified into four types: cultural tourists, circumstantial cultural tourists, cultural convenience tourists, and alternative tourists. One of the primary motivations for tourists to decide on a destination is the awareness of the heritage and culture of a place. In the other word, the design of a cultural tourism product for a city or region must meet the expectations of tourists in terms of heritage and cultural knowledge, as this is the primary motivation for the decision to choose a particular destination [40]. Moreover, tourists usually have high demands on the product, and therefore their interests and needs must be taken into account [1].

In general, there are enough products to satisfy users' consumer needs, so they need many in-depth cultural products to enrich their lives [31]. In this situation, a critical factor in developing good tourism policies for different destinations and seeking sustainable management models is to segment the various types of tourists and understand their motivations. However, little research has been conducted on the effect of the perceived value of souvenir consumption on affective attachment to destinations in cultural heritage tourism [17]. Therefore, to ensure that heritage is well preserved, it is necessary to study the factors influencing tourists' intention to purchase souvenirs [26], and also know the tourism capacity of the place, the socio-demographic situation of the tourists and their motivations, and also to deliver attractive tangible and intangible heritage to visitors and to enhance its value [1].

To sum up, perceived value is the most potent precondition for affective attachment to a destination in the context of cultural tourism. Thus, a better understanding of the experience of cultural heritage revival is beneficial for boosting the economic efficiency of tourism from the tourist's perspective [17]. From this theme, it can be inferred that proper recognition and dealing with consumer demand for cultural heritage products can positively affect the development of souvenir products. Future research may improve product competitiveness by further evaluating their relationship (Figure 7).

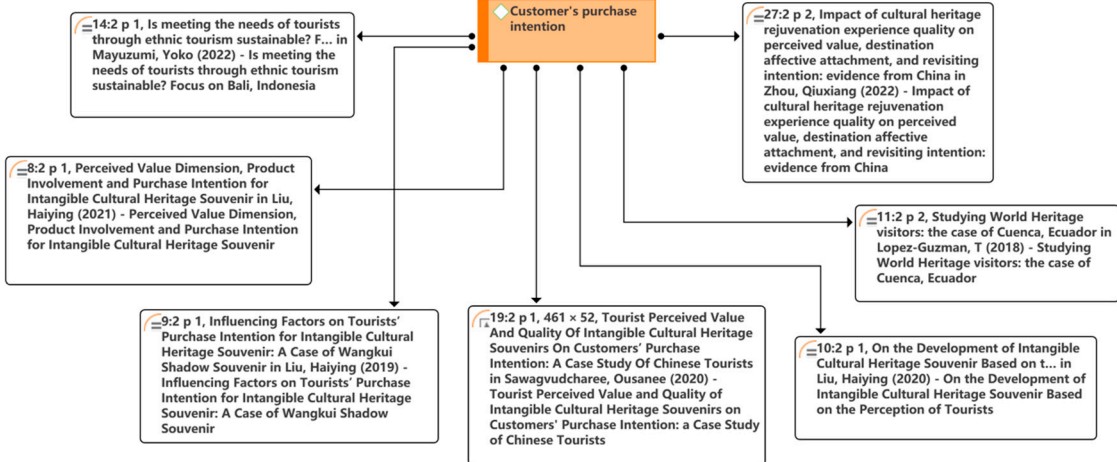

**Figure 7.** Network on the Customer's purchase intention [1,17,26,27,32,36,38].

4.2.3. Theme 3: Transformation

Ahmad [16] proposes that indigenous people's history, customs, and distinctive traditions can be used as tourism products. Historically, crafts were a significant income provider for some people, but the ensuing industrialization has led to a decline in interest in handicraft products. Design becomes a balancing instrument between product craftsmanship and industry [35]. Integrating intangible cultural heritage into modern design can provide rich design elements for tourism and innovative cultural products [12] (Figure 8).

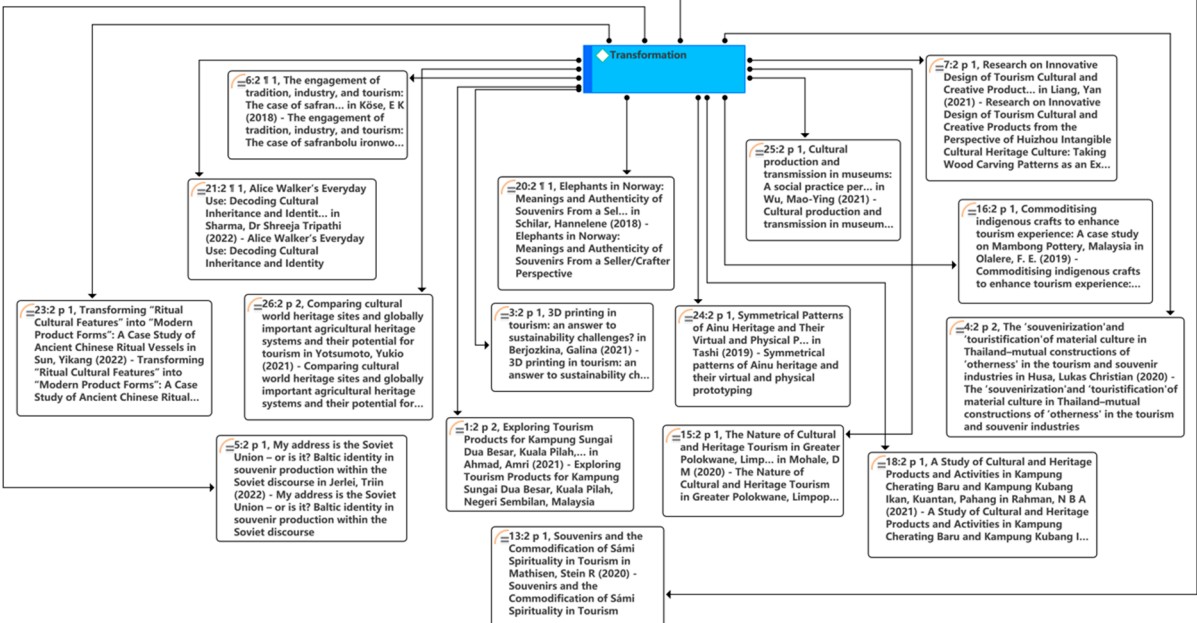

**Figure 8.** Network on the transformation [4,5,7,10–13,16,19,24,25,31,33–35,37].

Another case has shown that the Ainu in Japan often uses distinctive patterns to decorate their architecture, clothing, accessories, daily necessities, and spiritual objects [11]. Souvenirs and cultural artifacts made from these motifs, which carry their identity and aesthetic sense, are cherished by visitors, so they have cultural and commercial significance. A great deal of craftsmanship is required to reproduce these motifs accurately. However, there is a shortage of human resources with such skills at present and even for the foreseeable future. Therefore, there is an urgent need to preserve this craft, and digital fabrication technology can help solve this challenge of the pattern-making process.

However, 3D printing technology is more used to restore cultural sites in the tourism and hospitality industries, which has the potential in areas such as architecture, souvenirs, and food but has yet to be further fulfilled [33].

- Value transmission

The rapid development of the tourism market has impacted the transmission of the value of indigenous identity. This is because the growth of tourism and commercialization has allowed for series, mass production, and duplication, while artisans are more interested in preserving their skills [4]. On the other hand, product innovation makes it difficult to contextualize the meaning of historical symbols. However, there are always resources for innovation and transformation in indigenous arts and crafts and the invention of novel and interesting items to bring life to the production of souvenirs and effectively convey strong indigenous values.

Cultural resources include bundling or packaging products that can be made into tourist attractions using interpretation processes [37]. A conceptual cultural design model was used in an experimental study conducted by Olalere [5] to turn selected indigenous handicraft pottery into souvenirs that serve as a tangible medium to introduce cultural elements to tourists while conveying the spirit of the destination to potential visitors. This method is a feasible way to strengthen tourism and a sustainable solution to preserve local artisanship and generate opportunities for entrepreneurship. In his article, conversely, Schilar and Keskitalo [24] note that artisans and sellers are often reluctant to call their products souvenirs, believing that the word "souvenir" has a negative concept. However, they aware and agree that their products can become souvenirs during interactions with visitors. Respondents to this study further emphasized that storytelling has an essential role in the process of being or authenticating, and they voiced agency and authority over these statements.

Moreover, Sun et al. [31] propose using the "Ritual Vessels" of ancient Chinese ceremonial activities as prototypes for cultural product design and to convey the cultural essence to users through innovation and creative thinking. Similarly, Mathisen [4] points out that the main theoretical concentration of current research is how souvenirs can be adapted to general and Western tourism imagery, such as replicating the Sámi drum as a souvenir or applying symbols from the drum to souvenirs such as jewelry or other design products. Wu et al. [19] further propose bringing crafters and production processes to the forefront and bringing them into tourism events as a viable and innovative method of preserving and sharing craft heritage [19]. This method makes the manufacturing process transparent and allows users to engage in commercial activities and cultural dissemination practices, which is vital for enhancing craft heritage's cultural, social, and economic sustainability. Thus, the conventional method of transforming crafts into souvenirs might improve.

By using local resources as tourist attractions, visitors will be fascinated by what the destination offers and will instead visit these places further. In theory, any local resource can become a travel spot because tourists are interested in a wide variety of sights, and native communities need to identify, brighten and promote local features to appeal to visitors [25]. Therefore, "heritage" is frequently endowed with resources that can be turned into societal value to more fully recognize its value [41].

### 4.2.4. Theme 4: Sustainability (Figure 9)

Tourism is related directly to the prevalence, significance, and consequences of the sale and reproduction of ancient artifacts. At the same time, it stimulates the adoption of ancient motifs and styles of art that signal the feelings of the state and nation towards the heritage. As local artisans and artists can create tourist souvenirs that resemble "hidden treasures," there is a place for more professional souvenir marketing and production [9]. Tourism policy should encourage this development to achieve the goal of sustainable tourism economic growth [6].

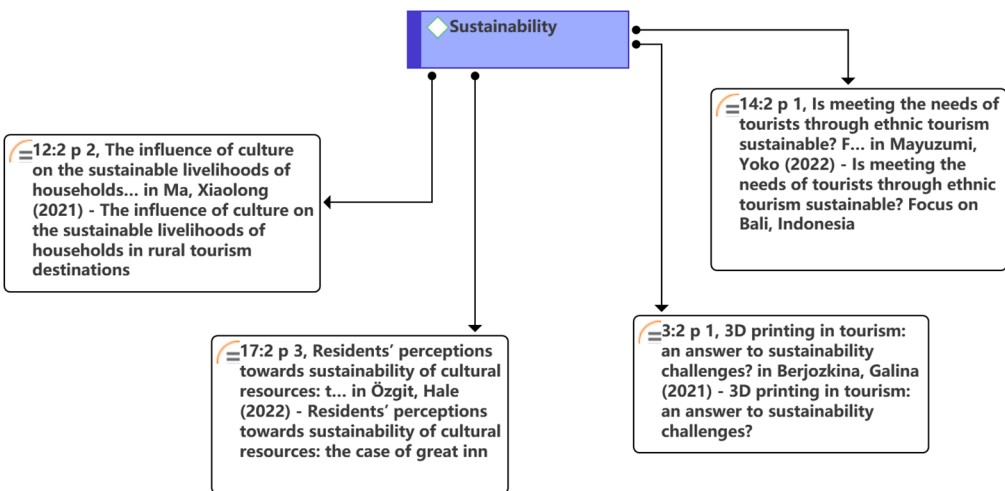

**Figure 9.** Network on the sustainability [20,22,27,33].

Traditional culture is a livelihood asset for families in some heritage sites, and it influences the thinking and behavior of local people. Areas with rich and unique cultural resources can raise awareness of cultural preservation among local communities and encourage their creative use of cultural resources to improve the living conditions of local people and develop sustainable tourism [20]. From a sustainable design perspective, 3D printing may be an effective tool for achieving the sustainable design of consumer goods. Three-dimensional technology significantly impacts sustainability regarding the material used and its potential to minimize waste and use less energy in tourism production [33].

The values of cultural heritage are currently used as marketing resources to attract tourists, and they can also generate tourism revenue [27]. Due to a town's cultural and historical sites having unique value, there is a definite trend towards ensuring the sustainability of cultural heritage, which means the increasing focus on cultural tourism further promotes cultural resources and sustainable development [22]. A finding from Mohale et al. [10] confirms that museums and art galleries create jobs and contribute to local economic development; however, the lack of resources has reduced their scale and operational capacity. Qiuxia et al. [42] suggest that product developers should satisfy consumers' demands, while destination culture can be embedded as inspiration for souvenir design. Innovation in design strategies will also enhance the efficiency of implementation, resulting in a scientific process for maximizing the benefit of souvenirs in terms of cultural and community sustainability. Therefore, a focus on the heritage sector and increased investment can facilitate the growth and sustainability of cultural and heritage tourism. Furthermore, to promote sustainable tourism development, it is fundamental to respect the socio-cultural authenticity of the host communities and preserve cultural heritage and traditional values [27].

## 5. A Conceptual Framework for Transforming Cultural Heritage into Souvenirs

Through the analysis of the articles and review of the research, a conceptual framework is presented that provides recommendations for new research. Figure 10 provides four main research directions to guide the use of cultural heritage resources in souvenirs, which can help identify new research opportunities, contribute to the strategic success of policymakers, and promote a hands-on approach to product design. The framework separately describes theories and concepts related to cultural heritage, tourists, and souvenirs, while constructing their logical relationship to sustainable development. The following categories are possible for future research given the current research trends.

1. Significance of souvenirs—to focus on souvenir value, role, and innovation related to cultural heritage.
2. Customer's purchase intention—to understand the relationship between consumer needs, perceived value, product innovation, and cultural heritage.
3. Transformation—to conduct in-depth research on souvenir production techniques and to focus on how to transmit value.
4. Sustainability—to create new values of cultural heritage in product development and strategy to better preserve cultural heritage in the premise of its commercialization.

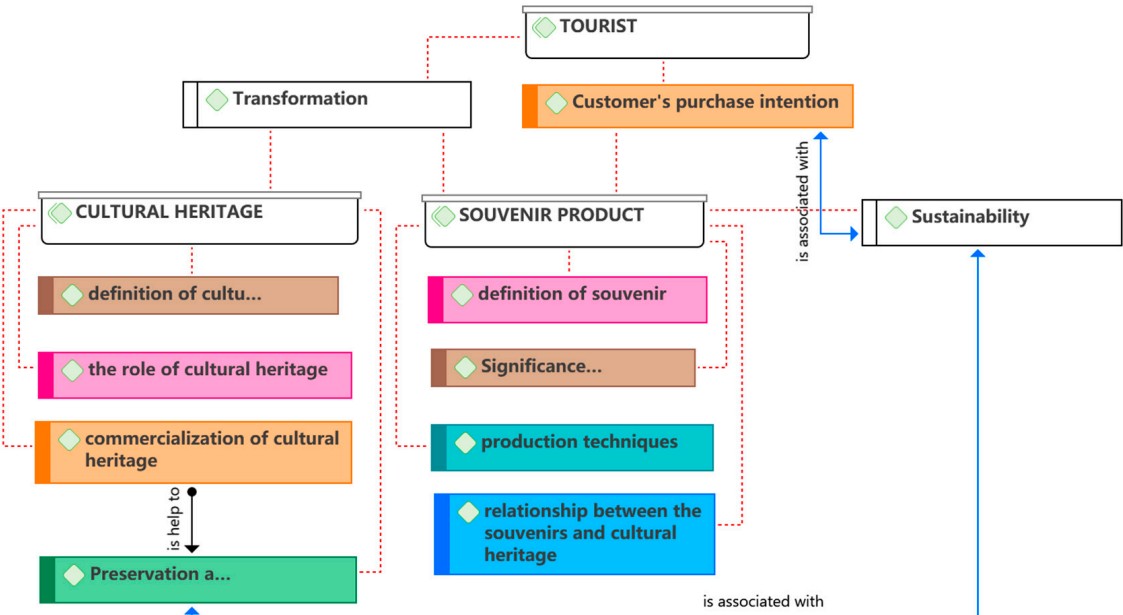

**Figure 10.** A conceptual framework for transforming cultural heritage into souvenirs.

## 6. Conclusions

This paper reviews 27 articles of the literature, focusing on cultural heritage-related souvenirs from 2018 to July 2022 to provide an overview of research and patterns on this topic. The ATLAS.ti 9 software was used to analyze the documents. First, the quantitative results revealed a lack of review articles reported in the literature that combine souvenirs and cultural heritage, despite the growing interest of researchers in this topic. Furthermore, linking the concepts addressed in the souvenir literature with cultural heritage and applying a more systematic terminology is necessary. Research on this topic has been slow in recent years, partly because policymakers and programmers are not fully aware of the role of theory in guiding the development of this industry. In the qualitative analysis section, key themes emerging from the literature that are relevant to the topic are highlighted. The relationship and the multiple factors that influence each other between cultural heritage and souvenirs are presented in this literature, thus predicting the direction of souvenir development. Some publications discuss the significance of souvenirs in the context of cultural tourism.

In contrast, others shed light on the relationship between consumer demand, products, and cultural heritage. Some studies focus on transforming cultural heritage resources into souvenirs and sustainability. The main contribution of this paper is to examine the literature on the association of souvenirs with cultural heritage. The practical contribution is to provide new directions for the industry's growth and to help the production and development of more culturally valuable products for industry personnel to maintain the industry's sustainability.

This paper highlights the need for research on creating and producing cultural heritage-related souvenirs to further enhance the cultural heritage value of products by explaining

the relationship between them. To souvenirize cultural heritage, it is recommended to clearly emphasize the significance of souvenirs, the features of cultural heritage, consumer needs, resource transformation methods, and techniques to complete the sustainability of the process.

**Author Contributions:** Methodology, Q.Z. and R.R.; software, Q.Z.; validation, H.A. and R.A.A.R.A.E.; resources, Q.Z.; writing—original draft preparation, Q.Z.; writing—review and editing, Q.Z.; visualization, Q.Z.; supervision, R.R., H.A. and R.A.A.R.A.E. All authors have read and agreed to the published version of the manuscript.

**Funding:** This research received no external funding.

**Institutional Review Board Statement:** Not applicable.

**Informed Consent Statement:** Not applicable.

**Data Availability Statement:** Not applicable.

**Acknowledgments:** The authors would like to thank Mohd Zairul for his help with the article composition.

**Conflicts of Interest:** The authors declare no conflict of interest.

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
