# Peer review of "Souvenirs Development Related to Cultural Heritage: A Thematic Review"

_sustainability, doi:10.3390/su15042918_

Round 1
Reviewer 1 Report
Review of the Manuscript “Souvenirs Development Related to Cultural Heritage: A Thematic Review”
The idea of applying IT achievements (namely, software ATLAS. Ti 9.) in the fields of humanitarian sciences (in particular, cultural studies) is certainly timely. Relation of souvenirs to cultural heritage is also of great scientific interest. According to the authors, they made review based on 27 articles devoted to this issue (souvenirs in relation to cultural heritage) and published in different journals (from 2018 to July 2022, and in three databases, which are WoS, Mendeley and Scopus). The paper is well illustrated with several figures and tables.
However, there are many repetitions in the paper. For instance, the authors constantly repeat the statement that souvenirs are not only a remembrance of the past, but also a marker of the culture. To my mind, it would be good to develop and deepen this thesis. In addition, I would like to point out some places in the article that need correction.
№ |
Page / Lines |
Comments |
1. |
Page 5. Lines 208–209. |
The sentence “The literature search was conducted in three Web of Science databases, Mendeley and SCOPUS” the word order is wrong. The word “databases” should be after the word “three”. |
2. |
Page 15. Lines 431–432. |
The sentence “However, their awareness and agreement that their products can become souvenirs during interactions with visitors” should be rewritten as “However, they aware and agree that their products can become souvenirs during interactions with visitors”. |
3. |
Page 18. Line 542. |
The phrase “Please add” seems redundant. |
4. |
Page 18. Line 546. |
In the sentence “The authors would like to thank her supervisor…” the word “authors” is in the plural form, but the word “her” is in singular form. |

Author Response
Thank you very much for taking your time to review my article.
Response to your Comments:
Point 1: The sentence “The literature search was conducted in three Web of Science databases, Mendeley and SCOPUS” the word order is wrong. The word “databases” should be after the word “three”. (Page 5. Lines 208–209.)
Response 1: Revised the sentence to “The literature search was conducted in three databases of Web of Science, Mendeley and SCOPUS”.
Point 2: The sentence “However, their awareness and agreement that their products can become souvenirs during interactions with visitors” should be rewritten as “However, they aware and agree that their products can become souvenirs during interactions with visitors”. (Page 15. Lines 431–432.)
Response 2: Revised the sentence to “However, they aware and agree that their products can become souvenirs during interactions with visitors”.
Point 3: The phrase “Please add” seems redundant. (Page 18. Line 542.)
Response 3: Deleted the words “Please add”.
Point 4: In the sentence “The authors would like to thank her supervisor…” the word “authors” is in the plural form, but the word “her” is in singular form. ( Page 18. Line 546.)
Response 4: Revised the word to “author”.
Point 5: However, there are many repetitions in the paper. For instance, the authors constantly repeat the statement that souvenirs are not only a remembrance of the past, but also a marker of the culture. To my mind, it would be good to develop and deepen this thesis. In addition, I would like to point out some places in the article that need correction.
Response 5: Removed one sentence to avoid repetitive statements about the role of souvenirs. ( Page 17. Line 519-521.)
Reviewer 2 Report
The paper treats an essential topic and reveals the authors' contributions to examining the literature on the association of souvenirs with cultural heritage. However, some minor revision is still necessary to enhance its profound value.
1. Authors included and conducted a thoroughgoing literature review in section 1. It could be more appropriate to separate the “Literature Review” from the “Introduction.”
2. Checking the use of brackets in Table 1 may be needed. Also, correcting the Arabic coding number for each Figure and Table will be necessary.
3. It’s significant to propose the conceptual framework for transforming cultural heritage into souvenirs. Moreover, how to enhance the importance of the “Significance of souvenirs ” with the consideration as same as other three emphases through the illustration of Figure 10 will be valuable.
Author Response
Response to Reviewer 2 Comments
Thank you very much for taking your time to review my article.
Point 1: Authors included and conducted a thoroughgoing literature review in section 1. It could be more appropriate to separate the “Literature Review” from the “Introduction.”
Response 1: Separated the “Literature Review” from the “Introduction" as the sencond section.
Point 2: Checking the use of brackets in Table 1 may be needed. Also, correcting the Arabic coding number for each Figure and Table will be necessary.
Response 2: Corrected the use of brackets in Table 1. Corrected the Arabic coding number for each Figure and Table.
Point 3: It’s significant to propose the conceptual framework for transforming cultural heritage into souvenirs. Moreover, how to enhance the importance of the “Significance of souvenirs ” with the consideration as same as other three emphases through the illustration of Figure 10 will be valuable.
Response 3: Add more explanation of the composition and details of the conceptual framework. (Page 16. Lines 484–496.)
Reviewer 3 Report
The introduction clearly presents the methods, typology and aims of the research carried out. The issue addressed concerning the pairing of souvenirs and cultural heritage is very interesting, as are the criteria used in the choice of published material at the basis of the research.
Overall, the contribution is structured in such a way as to provide an organic view of the subject matter, with an exposition in very understandable English and with punctual and pertinent references.
The conclusions adequately summarize the research and the results achieved, paving the way for future studies on creating and producing cultural heritage-related souvenirs.
It is recommended the publication in the present form.
Author Response
Thank you very much for taking your time to review my article.
Point: It is recommended the publication in the present form.
Response: No revision available.
Round 2
Reviewer 1 Report
My comments have been taken into account.
However, I would like to draw attention to this point once more, as the manuscript has more than one author.
"Point 4: In the sentence “The authors would like to thank her supervisor…” the word “authors” is in the plural form, but the word “her” is in singular form. ( Page 18. Line 546.)
Response 4: Revised the word to “author”".
Author Response
Thanks for your time.
Point 4: I would like to draw attention to this point once more, as the manuscript has more than one author.
Response 4: Corrected the grammar of this sentence: The authors thank Dr. Mohd Zairul for his help with the article composition. ( Page 18. Line 550-551.)
